# Identification of Novel Glycans in the Mucus Layer of Shark and Skate Skin

**DOI:** 10.3390/ijms241814331

**Published:** 2023-09-20

**Authors:** Etty Bachar-Wikstrom, Kristina A. Thomsson, Carina Sihlbom, Lisa Abbo, Haitham Tartor, Sara K. Lindén, Jakob D. Wikstrom

**Affiliations:** 1Dermatology and Venereology Division, Department of Medicine (Solna), Karolinska Institutet, 17177 Stockholm, Sweden; 2Whitman Center, Marine Biological Laboratory, Woods Hole, MA 02543, USA; 3Proteomics Core Facility of Sahlgrenska Academy, University of Gothenburg, 41390 Gothenburg, Sweden; 4Department of Fish Health and Welfare, Norwegian Veterinary Institute, P.O. Box 750, Sentrum, 0106 Oslo, Norway; 5Department of Medical Biochemistry and Cell Biology, Institute of Biomedicine, Sahlgrenska Academy, University of Gothenburg, P.O. Box 440, Medicinaregatan 9C, 40530 Gothenburg, Sweden; 6Dermato-Venereology Clinic, Karolinska University Hospital, 17176 Stockholm, Sweden

**Keywords:** elasmobranchs, sharks, skin, mucus layer, mucin, glycans, glycoproteins, *O*-glycans, *N*-glycans, mass spectrometry

## Abstract

The mucus layer covering the skin of fish has several roles, including protection against pathogens and mechanical damage. While the mucus layers of various bony fish species have been investigated, the composition and glycan profiles of shark skin mucus remain relatively unexplored. In this pilot study, we aimed to explore the structure and composition of shark skin mucus through histological analysis and glycan profiling. Histological examination of skin samples from Atlantic spiny dogfish (*Squalus acanthias*) sharks and chain catsharks (*Scyliorhinus retifer*) revealed distinct mucin-producing cells and a mucus layer, indicating the presence of a functional mucus layer similar to bony fish mucus albeit thinner. Glycan profiling using liquid chromatography–electrospray ionization tandem mass spectrometry unveiled a diverse repertoire of mostly *O*-glycans in the mucus of the two sharks as well as little skate (*Leucoraja erinacea*). Elasmobranch glycans differ significantly from bony fish, especially in being more sulfated, and some bear resemblance to human glycans, such as gastric mucin *O*-glycans and H blood group-type glycans. This study contributes to the concept of shark skin having unique properties and provides a foundation for further research into the functional roles and potential biomedical implications of shark skin mucus glycans.

## 1. Introduction

Elasmobranchs, including sharks, have received a great deal of attention in research due to conservation efforts, but their molecular biology is also of great interest despite the challenges associated with experimentation. Previous studies have led to several significant discoveries with potential applications in human medicine, such as the identification of the antibiotic squalamine [1] in the liver and stomach of spiny dogfish sharks and research on chloride channels in the rectal gland of these sharks [2], which are relevant to cystic fibrosis.

Fish skin shares several structural similarities with mammalian skin. It consists of three epidermal layers, with the outermost layer being the stratum superficiale composed of differentiated cells. The next layer is the stratum spinosum, which contains differentiating cells, and the innermost layer is the stratum basale, which includes proliferating basal cells and a basement membrane that borders the dermis [3,4]. Depending on factors such as the fish species, age, location on the body, thickness of the epidermis, and number of epidermal layers, various specialized cells may be present in the epidermis. These cells can include mucin-producing goblet, sacciform, and club cells as well as alarm cells and chloride cells in addition to keratocytes, which are the fish equivalent of mammalian keratinocytes [5].

One key difference between fish and mammalian skin is that almost all fish species lack the dead, keratinized protective layer known as the stratum corneum. Instead, the fish epidermis consists entirely of living cells [6,7] and is protected by a layer of mucus, a slimy substance composed mainly of a high molecular weight and heavily glycosylated proteins referred to as “mucins”, which are important for mucus viscosity, trapping pathogens, physically protecting the skin surface, and contributing to signaling at the cell surface [8]. Many mucins cross-link in solution via disulfide bonding and this can promote the formation of a gel-like substance. In addition, there are also smaller less glycosylated proteins, some of which have antimicrobial properties that help prevent the entry and establishment of pathogens [6,9]. Thus, while almost all secreted proteins are glycosylated [10], the extent of glycosylation and molecular weights varies. The mucins, which give the mucus its viscous, elastic, and adhesive properties, are proteins post-translationally modified with monosaccharides attached with glycosidic bonds. On mucins, these glycans are mostly in the form of *O*-glycans that attach to oxygen atoms on serine or threonine in proteins; however, *N*-glycosylation also occurs [11]. *O*-glycans are more common and can be further classified into core types 1–8. Core 1 is composed of a galactose and is attached to the base *N*-Acetylgalactosamine (GalNAc), while core 2 utilizes the core 1 complex with an addition of *N*-Acetylglucosamine (GlcNAc) to the GalNAc, and cores 3–8 are synthesized in a similar way [12,13,14]. Glycan biosynthesis take place in the endoplasmic reticulum and Golgi organelles and is performed via glycosyltransferases.

Although the mucus layer glycomes of some common bony fish (*Osteichthyes*) grown in aquaculture are well described [15,16], little is known about the glycomes of elasmobranchs, including sharks. Shark skin possesses unique features, such as its teeth-like denticles, and it is possible that the mucus layer may also have distinct properties and functions, such as providing defense against pathogens. Investigation into the composition and function of the mucus layer in sharks could lead to valuable insights and potential applications in various fields. A molecular characterization of shark mucus is an essential first step towards understanding its biology. Using liquid chromatography–electrospray ionization tandem mass spectrometry, this pilot study presents the most comprehensive description of shark mucin glycosylation to date in Atlantic spiny dogfish (*Squalus acanthias*), one of the most common shark species, and compares it to chain catsharks (*Scyliorhinus retifer*) as well as little skates (*Leucoraja erinacea*).

## 2. Results

### 2.1. Shark Skin Histology

Regular bony fish (*Osteichthyes*) have scales, while the so-called placoid scales of chondrichthyans and specifically elasmobranchs are described as tooth-like denticles due to their outer enamel covering, a dentine layer, and an inner pulp cavity [17] as well as that teeth and denticles both continuously renew and share developmental and genetic similarities as shown in small-spotted catsharks (*scyliorhinus canicular*) [18]. To establish the basic histology of our shark species’ skin (Figure 1) we first performed hematoxylin-eosin (HE) staining of skin biopsy tissue and also Masson’s trichrome (MT) staining to identify collagen as well as keratin and muscle fibers as described before [19] (Figure 2). Of note, chain catshark skin was much tougher and significantly more difficult to penetrate with the biopsy tool, perhaps due to different biological needs, as catshark females are often injured by males during mating ([20] and personal observation). Figure 2A shows that spiny dogfish and chain catsharks are covered by a skin scattered with dermal bony denticles that differ histologically, as spiny dogfish have backward-pointing spine-like denticles, while catsharks have narrow and flat hammer-like denticles, as well as a dermal bony basal plate as previously described [19,21].

As few previous studies have examined mucin-producing cells in elasmobranchs, we then performed periodic acid–Schiff (PAS) staining. Indeed, although sharks do not appear “slimy” when handled compared to bony fish such as salmonoids (own personal observation), there were plenty of mucin-positive cells that appeared as empty white goblet cells in the HE sections (Figure 3 and Figure 4A) and pink goblet cells in the PAS sections (Figure 3 and Figure 4B). Furthermore, there was a pinkish staining on the skin surface indicative of a mucus layer (Figure 3D,H and Figure 4B,F).

### 2.2. Glycan Mass Spectrometry

Glycans were analyzed with LC-MS. Since the samples had a low abundancy with respect to the mucin-type *O*-glycans, they had to be pooled.

#### 2.2.1. Spiny Dogfish

Spiny dogfish samples were pooled into six pools (2–3 individuals per LC/MS sample). The LC/MS data from these six pooled samples did resemble the initial pilot experiment with the same glycans, but they were quite weak, one sample was empty, and the remaining five contained between 3 and 15 glycans (Appendix A).

To obtain better structural data, and to generate two separate analyses with maximum glycan coverage, leftovers from the six pooled samples were pooled and reanalyzed with LC/MS (Figure 5), as well as the sample from the pilot experiment, using both MS^2^ and MS^3^ experiments, to obtain more information for structure assignment. The data were compiled in a spreadsheet (Appendix A). 

In all, the glycan profiles in all pooled samples resembled each other, revealing 39–40 *O*-glycans in the size range of 2–9 residues (Figure 5 and Appendix A). The glycans were mainly neutral. We detected only low levels of the short NeuAc (*N*-acetyl neuraminic acid)-containing glycans (675a and 675b) (1.0%) (Appendix A). No traces of other acidic glycans containing *N*-glycolyl neuraminic acid (NeuGc) or diamino neuraminic acid (KDn) were detected. One sulfated glycan was detected (*m*/*z* 464 at 8.3 min, Figure 5A and Appendix A (0.2%)). Since the previously characterized fish glycomes mainly contained acidic glycans [14,15,16,22], this made us concerned about how representative the glycans identified were. Therefore, we analyzed skin sections using PAS/AB staining, which detects both sialylated and sulfated glycans and stains the skin goblet cells of the teleosts previously analyzed, mainly in blue, due to their high content of acidic mucins [23]. The absence of blue goblet cells via PAS/AB stain confirmed the low abundance of acidic glycans (Appendix A).

The glycans were mainly core 1- and core 2-type glycans, which are also found on mammalian and salmon mucins [12,22]. The majority (81.2%) were fucose (Fuc), a deoxyhexose that is present on polysaccharides- containing glycans. A major component was a glycan at *m*/*z* 530 (deHex-Hex-HexNAcol, Figure 5A,B 27.5 min), which had the same MS^2^ spectra as the blood group H glycan from human saliva (Fuc(α1-2)Gal(β1-3)GalNAcol (Figure 5C). This is a common glycan on mucosal secretions in mammalians, but Fuc-Gal- is absent on bony fish, such as zebrafish, Atlantic salmon, Arctic char, and rainbow trout, where Fuc is only found linked to HexNAc’s (GlcNAc or GalNAc). One glycan with Fuc linked to HexNAc was observed here (*m*/*z* 1187 at 28.9 min) (Appendix A). Figure 5D shows a glycan 1041b, which we have interpreted as a core 2-type *O*-glycan with two blood group H epitopes (Fucα1-2Galβ1-3[Fucα1-2Galβ1-3GlcNAcβ1-6]GalNAc. In mammalian secretions, both type 1 (Galβ1-3GalNAc-) and type 2 (Galβ1-4GalNAc-) chains make up the extended branches. Compared to published reference spectra, the lack of a fragment ion at *m*/*z* 409 supports that this glycan is of a type 1 chain (Figure 5D). 

MS^2^ spectra of glycans at *m*/*z* 790b and *m*/*z* 936 (Figure 5A) were found to be similar to published spectra of extended core 5 glycans (GalNAcα1-3GalNAc), which are present on mucosal surfaces of Atlantic salmon and rainbow trout but not in mammalian systems [14].

Two glycans with three adjacent HexNAc’s (polyHexNAc) in a row were observed at *m*/*z* 790 (790a, Appendix A) and 993 (Figure 5E, Appendix A). This is not a common monosaccharide sequence among the mammalian or fish glycans published so far. The glycan at *m*/*z* 790 with a Hexose as the core (HexNAc-HexNAc-HexNAc-Hexol, Appendix A) may be a degradation product since the *O*-glycan core residue (linked to the protein) is commonly a HexNAc (GalNAc).

Nine glycans displayed MS^2^ spectra resembling those of mammalian *N*-glycans (3–7%, Appendix A). It is not unusual to observe *N*-glycans in mucosal secretions, since these are also released during the beta-elimination reaction. The *N*-glycans were of a predominantly high-mannose type, as well as one complex *N*-glycan and two truncated *N*-glycans of the pauci-mannose type.

**Figure 5 ijms-24-14331-f005:**
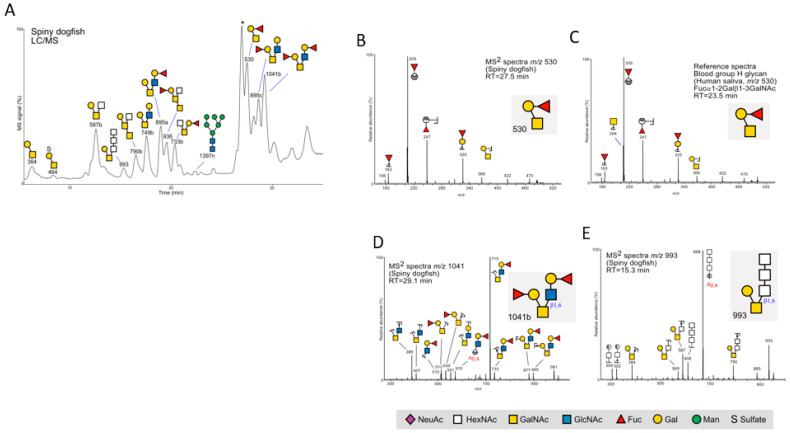
Spiny dogfish glycans. (**A**) Pooled glycans analyzed in their reduced nonderivatized form using LC/MS in the negative ion mode. The most abundant glycans are annotated using the SNFG nomenclature [24]. * = nonglycan contaminant. (**B**) Spiny dogfish skin secretions contain glycoproteins carrying human blood group H-type epitopes. MS^2^ spectra of a glycan detected at *m*/*z* 530 from the skin of spiny dogfish and (**C**) MS^2^ spectra of the blood group H glycan detected at *m*/*z* 530 ([M-H]^-^ precursor ion) of *O*-glycans from human salivary glycoproteins, analyzed at the same occasion. (**D**)Additional glycans were detected in spiny dogfish. MS^2^ spectra of a glycan with blood group H-type epitopes detected at *m*/*z* 1041 eluting at 29.1 min (1041b,(**A**)) (**E**) MS^2^ spectra of an *O*-glycan detected at *m*/*z* 993. Diagnostic cross-ring fragments (^0,2^A_0_) are annotated in red.

#### 2.2.2. Chain Catshark

Chain catshark samples were pooled into two samples from males and three from females; however, hardly any glycans were detected in these five samples. Therefore, we pooled all samples into one and reanalyzed it, allowing the detection of two glycoforms at *m*/*z* 749 with the same residue configuration and one glycan at *m*/*z* 895 (Figure 6). The two 749 glycoforms may arise from the type 1 and 2 linkage glycoforms (Galβ1-3GlcNAc and Galβ1-4GlcNAc), which, as mentioned above, are found on mammalian mucins. 

#### 2.2.3. Little Skate

In order to extend the analysis beyond sharks to skates, another main group of the elasmobranch family, we obtained four skin mucus samples from little skates. The BCA protein assay revealed low overall protein amounts, so these samples were divided into two pools and analyzed with LC/MS and MS^2^. We detected 14 and 16 *O*-glycans, respectively, in the two samples (Appendix A). Since the spectra were relatively weak, we pooled these two samples and then detected 22 *O*-glycans (Appendix A).

After the first analyses, the two samples were pooled and analyzed again using MS^3^. The glycans consisted of 2–6 residues and were of core 1 and 2 types. Similarly to the glycans detected in spiny dogfish, Fuc (relative abundance 91.5%) was found only linked to Hex residues (Appendix A).

Little skate *O*-glycans contained relatively more acidic glycans than spiny dogfish, as they contained sulfate groups and NeuAc, constituting 26.1% of the total glycan peak area. Four and nine of the glycans detected contained NeuAc and sulfate (26.1% and 41.7%, respectively, Appendix A). These acidic residues are common on mammalian and fish glycans [14,25,26], although the arrangement with sulfate and Fuc linked to the same Hex residue has not been described before (Figure 7). Figure 7B shows the MS^2^ spectra of the glycan interpreted as Fuc-(SO_3_^−^)Gal-[HexNAc-]GalNAc, eluting at 18.2 min and detected at *m*/*z* 813 ([M-H]^-^ precursor ion, panel B-1). MS^3^ of the fragment ion at *m*/*z* 610 (M-HexNAc) (panel B-2) reveals the presence of a B-ion at *m*/*z* 387, composed of sulfate, a Hex, and a Fuc residue, and in the MS^3^ spectra of the fragment ion at *m*/*z* 667 (M-Fuc), the fragment ion at *m*/*z* 241 is diagnostic for sulfate linked to the Hex residue (panel B-3).

In one of the three samples, three *N*-glycans with a low abudance were detected, making up 2.5% of the total glycan pool. The double-charged glycan detected at *m*/*z* 860.3 eluting at 21.0 min was interpreted as a sulfated *N*-glycan, which is rarely found in mammals. The MS^2^ spectra are shown in Figure 7C. In addition to typical *N*-glycan fragment ions generated by the loss of residues from both the reducing and the nonreducing end, a major diagnostic ion at *m*/*z* 1623.6 (Figure 7C) was interpreted as a loss of sulfate and H_2_O, in addition to fragment ions in the lower mass range at *m*/*z* 241 and 444, supportive of sulfated Hex and HexHexNAc, respectively. Although sulfated *O*-glycans are commonly present on mucosal proteins, it should be noted that low-resolution MS, which was applied here, does not provide the mass accuracy to distinguish between sulfate and phosphate (79.9568 and 79.9799 amu, respectively).

## 3. Discussion

In this study, we provide a detailed characterization of the skin of three types of elasmobranchs, with a particular focus on mucin glycosylation. Our findings highlight several key points, including the presence of numerous secretory cells in the skin and the identification of several unique glycans in sharks. In the following sections, we discuss these findings in detail.

### 3.1. The Mucus Layer in Bony Fish and Elasmobranchs

The mucus layer in bony fish is a dynamic and complex mixture of various molecules that serve both protective and immunological functions. The most important components of the mucus layer are mucins, which are large glycoproteins that provide viscosity and adhesion to the mucus layer and also act as a physical barrier to pathogens [27]. The mucus layer also consists of numerous other molecular components, such as antimicrobial peptides, immunoglobulins, complement proteins, lysozyme, and lectins, that directly attack or neutralize invading pathogens and is created through secretion as well as sloughing of dead cells [28]. Additionally, the mucus layer also contains various cytokines and chemokines that attract immune cells to the site of infection, promoting a local immune response. While many antimicrobial peptides have been identified in fish mucus and explored for their potential therapeutic use in humans, only a few have been studied in clinical trials. An extensive list of antimicrobial agents found in teleost is nicely presented in a review [29].

Elasmobranchs (sharks, skates, rays, and sawfishes) are among the oldest and most diverse marine vertebrates and differ from bony fish in several aspects, including a cartilaginous skeleton and that the skin is covered by placoid scales (denticles) that reduce fluid friction and thus enhance swimming efficiency [30]. The mucus layer of elasmobranchs is far less researched than in bony fish and the specific components are unknown, although it is likely different to bony fish due to dissimilar skin architectures as well as extensive evolutionary separation. Although not formally tested in any studies to our knowledge, it is believed that elasmobranchs only have a thin mucus layer [31]. No studies to our knowledge have examined glycans in elasmobranchs; however, a study on Japanese bullhead sharks (*Heterodontus japonicus*) identified a C-type lectin that belongs to a group of carbohydrate-binding proteins [32]. Moreover, several studies have mapped the skin bacterial microbiota of sharks and rays [33], and several antibiotic-producing bacteria were identified on skate skin [34].

### 3.2. Shark Skin Histology

Several studies have demonstrated that the skin of sharks possesses secretory cell types, despite not exhibiting a slimy texture upon handling (e.g., *Scyliorhinus canicular*, small-spotted catshark) [31]. These cells can be found stretching from the stratum basale to the epidermal surface and may be similar to the epidermal secretory cells of bony fish. With this in mind, we analyzed skin biopsy sections from spiny dogfish and chain catsharks and found a significant presence of secretory cells throughout the epidermis. These voluminous cells are likely columnar goblet cells or granular cells [31]. In comparison to rays, small-spotted catsharks display a higher abundance of secretory cells, with 40 secretory cells per 100 basal cells observed in the dorsal region and 20 secretory cells in the front region and fins. This is in contrast to nurse sharks (*Ginglymostoma cirratum*), which exhibit a much lower number of voluminous secretory cells [31]. 

In our work, PAS staining unveiled distinct secretory cell coloration differences between the shark species. In skin samples obtained from dogfish, secretory cells, presumably of the goblet type, were magenta-pink in color upon PAS staining. Conversely, in catsharks secretory cells displayed magenta-pink coloring as well as light green staining. The differential staining observed between the two shark species may be attributed to the use of a light green counterstain in conjunction with PAS staining. This counterstain is utilized to enhance tissue definition and highlight glycogens and mucins, which can result in a PAS light green/blueish color [35]. Similarly, in biopsies of the mucosa-rich human esophagus stained with PAS/AB, goblet cells containing acidic mucin-filled vacuoles can distend the cytoplasm and result in a blue discoloration rather than the typical pink hue [36]. Furthermore, certain images depict the presence of pink staining within the blueish secretory vacuole, suggesting the co-existence of distinct mucins within the same secretory vacuole (Figure 2B,D,F). Thus, the histological findings indicate that the mucin in catsharks may possess a dissimilar chemical structure, potentially exhibiting a greater acidity, in comparison to the mucin present in dogfish. Overall, our histology data establishes the presence of a mucus layer in the examined shark species as well as adding to and complementing the limited histology literature on shark skin.

### 3.3. Glycans Unique to Elasmobranchs and Potential Roles

The mucins, making up the majority of the mucus layer, are highly glycosylated with the glycans often contributing 50–80% to the molecular weight of the glycoconjugate [37]. There are several glycan sugar modifications, including sialylation, sulfation, and fucosylation, with important regulatory functions, for example, mucosal immune functions [38]. The *O*-glycans can protect against pathogens, by adhering to bacteria and acting as releasable decoys [39], thereby preventing them from interacting with the epithelial cells [22]. Moreover, *O*-glycans are hydrophilic and usually negatively charged when present on fish skin [14,15,16,22]; they promote the binding of water and salts and are major contributors to the viscosity and adhesiveness of mucus, which forms a physical barrier between the surrounding water and epithelium [11].

In this study, we characterized *O*-glycans and *N*-glycans into three types of elasmobranchs, the first such mapping performed. In spiny dogfish and little skates, we detected 39 and 22 glycans, respectively, mainly core 1 and core 2. In dogfish, the glycans were mostly neutral and not acidic (heavily sialylated or sulfated), and the majority were *O*-glycans, yet a few *N*-glycans were also found. In skates, we found more acidic glycans, all *O*-glycans. In both species, most of the glycans were heavily fucosylated and to a lesser extent exhibited a HexNAc termination. The mucus harvested from catsharks was not sufficiently concentrated to allow a complete glycan extraction, although three glycans with Galβ1-3/4GlcNAc repetition were found.

In dogfish, two glycans with three adjacent HexNAc’s (polyHexNAc) were found, which is rare. This sequence is a carbohydrate polymer composed of *N*-acetylhexosamine residues, which was first isolated from bacteria [40]. PolyHexNAc was later found in heartworms (*dirofilaria immitis*) and showed a strong immunoactivity [41], as well as in fungi where it serves as an antioxidant [42]. It is plausible to think that the polyHexNAc glycans were derived from the microbiota community on the sharks if not from the sharks themselves; however, it is unlikely because of the relatively high abundance of these glycans and the fact that no large amounts of bacteria were detected (Appendix A).

When examining the dogfish glycans in depth, we discovered, to our surprise, that the *O*-glycans in dogfish show an outstanding resemblance to human gastric mucin *O*-glycans. Rossez et al. demonstrated that the *O*-glycans residing in human gastric mucus are, as in dogfish, mostly neutral, highly fucosylated, and carry several lactosaminic units (repetition of Galβ1-3/4GlcNAc) [43]. Core 2 was the main core structure detected in these gastric mucins, and numerous fucosylated oligosaccharides carried the blood group O determinant (Fucα1-2Galβ1). Moreover, they showed that these glycans can serve as potential binding sites for bacteria such as *Helicobacter pylori* [43]. The ability of fish mucins to bind bacteria is also a known concept for fish mucus, as demonstrated in salmonoids [27,44,45]. This similarity between two genetically remote species (human and shark) suggests the following: 1. Some glycans may be very conserved evolutionarily and have similar functions. 2. Shark biology may have relevance to humans with potential medical translational applications.

As mentioned above, the fucosylation modification on the dogfish (and skate) glycans was very abundant. Indeed, in Atlantic salmon and zebrafish fucosylation *N-* and *O*-glycans are common [14,46]. However, the arrangement of the Fuc group within the glycan is different in dogfish (Figure 3). Fucose mediates protein interactions, which are essential to biological processes, such as host–microbiota communication, viral infection, or immunity [47]. In immunoglobulins, core fucosylation is particularly important. For example, the fucosylation of IgG antibodies shifts the balance of Type I and Type II Fc gamma receptors (FcγR) that will be engaged by immune complexes, which, in turn, modulates the effector cells and functions that can be recruited during immune activation [48]. In mammals, fucosylation is highly important as it constitutes a component of the ABO blood group. Interestingly, we found that both spiny dogfish and little skate skin secretions contain glycoproteins carrying the human H blood group-type glycan. The H blood group type 1 glycan epitope, Fucα1–2Galβ1–3GlcNAc, is expressed at the termini of *O*-glycans on a high-molecular-weight sialomucin and at the nonreducing termini of *N*-glycans on a number of unidentified glycoproteins of a medium molecular size [49]. In humans, this epitope is encoded by either FUT1 (fucosyltransferase 1, in blood) or FUT2 (epithelial cells on mucosal surfaces), which is required for the final step of the synthesis of soluble A and B antigens [50], but, interestingly, also mediates diverse biologic processes, such as angiogenesis, macrophage polarization, keratinocyte migration^,,^ and cancer cell survival [51,52]. Its role in fish and sharks is currently unknown. 

The skin mucin *O*-glycomes of Atlantic salmon (*Salmo salar*) and rainbow trout (*Oncorhynchus mykiss*) have previously been described in detail [14,15,16,22]. The sample collection and mucin isolation process used differed from the current study, but the sample preparation and MS were performed using the same methodology. The glycans detected in the elasmobranchs in the current study were notably different from the glycans detected in the salmonids (Figure 8). Since the current study was based on less material, we compared the glycans making up >50% of the glycans. Although 20–60 glycans (depending on species) have been detected on the salmonid skin mucins, the vast majority of the glycans are short (two monosaccharides) and acidic, with the acidic moiety mainly being comprised of sialic acids instead of the sulfation detected in the current study (Figure 8). Low levels of sulfation are also found in the salmonids; however, sialylation dominates by far. The salmonid mucin’s short and sialylated glycans have a poor pathogen binding ability, possibly due to steric hindrance/too short epitopes, since fish pathogens bind larger sialylated epitopes on other epithelial sites in these salmonids with higher avidity [15,44]. Indeed, we have speculated that the short glycans on the salmonid skin glycans act akin to Teflon, to limit the number of bacteria attaching to the external surface of the fish, in contrast to the mucins produced on internal epithelial sites that appear to act as releasable decoys, transporting pathogens away from the epithelial surface. Medium-sized fucosylated glycans, similar to those dominating on the elasmobranch surface, are mainly found on the gills in the salmonids and to a lower extent in the gastrointestinal tract. Since mucins from these regions in the salmonids have a higher avidity for pathogen binding than the skin mucins, one may speculate that the elasmobranch skin mucins bind bacterial pathogens efficiently and have a different function/role with regards to interactions with bacteria than the salmonid mucins, possibly providing nutrients for a beneficial microflora [53].

### 3.4. N-Glycans

We discovered three *N*-glycans in dogfish. It is well known that *N*-glycans are important in retaining growth factor and cytokine receptors at the cell surface, probably through interactions with galectins or cytokines such as TGF-β [54]. It is reasonable to assume that the low number of *N*-glycans in dogfish and their absence in the other two types of sharks is either true since *N*-glycans are less common than *O*-glycans in the bony fish mucus layer or false due to the method used to collect the mucus layer. However, supporting the lower abundance of *N*-glycans compared to *O*-glycans is that if present they are found on most proteins in mucosal layers and are thus easily detected [55].

### 3.5. Study Limitations

The spiny dogfish were only females. The mucus harvest method may have missed some glycans and did not work well in chain catsharks in which longer absorption times or scraping may be needed. 

## 4. Materials and Methods

### 4.1. Animals

Spiny dogfish caught by hook gear were purchased from commercial fisherman in Chatham, MA, in 2022. Only female spiny dogfish were available, likely due to commercial fishing often targeting female schools [56]. Chain catsharks were collected from a National Oceanic and Atmospheric Administration survey vessel by dredging in the mid-north Atlantic between 2017 and 2019. Skates were collected via trawl net around Woods Hole, MA, by the Marine Biological Laboratory (MBL) in 2021. All elasmobranchs were housed in tanks with natural sea water flow-through systems, which were maintained year-round at 14 °C at the Marine Resources Center (MRC) at the MBL. Elasmobranchs are housed in single-species groups, and they are fed a diet of food-grade frozen capelin (Atlantic-Pacific North Kingstown, RI) and fresh frozen locally caught squid three days per week. Photos were taken with an iPhone 13 Pro (Apple Inc. Cupertino, CA, USA). Experiments were approved by the Institutional Animal Care and Use Committee (IACUC) at the MBL (protocol no 22-22).

### 4.2. Skin Mucus Sampling

Skin mucus was sampled using the Kleenex tissue absorption method previously developed for salmonoids [57]. Briefly, housed elasmobranchs were caught gently with a net and a Kleenex tissue was placed on the skin for 10 s to saturate it with mucus fluid before it was put in the upper compartment of Spin-X tubes (Sigma-Aldrich, St. Louis, MO, USA ) on ice and later spun down at 700 g in a 4 °C cooled benchtop centrifuge to collect the absorbed mucus fractions. Tank water controls samples were also harvested by placing the Kleenex (Kimberly-Clark, Irving, TX, USA) briefly in the tank water. The liquid samples were transferred to plastic cryotubes, snap frozen on dry ice, and stored at −80 °C. The samples had volumes of 0.6–1 mL. The sample protein content was estimated via a bicinchoninic (BCA) assay (Thermo-Fisher, Whaltham, MA, USA).

### 4.3. Skin Biopsy Sampling and Histology

For skin biopsies, the elasmobranchs were gently caught with a net and transferred to a plastic procedure tank of 90 l (cooler style, with a lid) with the general anesthetic AQUI-S^®^ 20E (eugenol) under INAD #11-741 37.5 mg/L dissolved in sea water. The animals were maintained under anesthesia via a water pump delivering anesthetic sea water into the mouth. Biopsies were harvested using 4 mm punch biopsy tools (Kai Medical, Solingen, Germany) and fixed in 4% formaldehyde followed by paraffin embedding and sectioning. Staining was performed at the ZooQuatic Laboratory (Durham, NH, USA), according to standard protocols.

### 4.4. Mass Spectrometry

Glycans were released from the proteins and analyzed in their reduced form as nonderivatized alditols with liquid chromatography connected to mass spectrometry kept in the negative ion mode and sequenced using collision-induced dissociation (CID) in MS^2^ and MS^3^ experiments. 

#### 4.4.1. Glycan Release

Samples were analyzed using a standard glycomics workflow as described below.

The method for glycoprotein dot blot, glycan release, and analysis used here is described in detail elsewhere [58]. The method is optimal for *O*- and *N*-glycans consisting of 2–16 monosaccharide residues.

Briefly, the shark skin samples were dried down using a speedVac vacuum concentrator (Thermo-Fisher), then the proteins were reduced in 400 µL of extraction buffer (0.1 M dithiothreitol, ultrapure 6 M guanidinium hydrochloride (MP Biomedicals, Santa Ana, CA, USA), 5 mM EDTA, 0.1 M triethylamine bicarbonate buffer; pH 8.1) and kept at 37 °C overnight. The samples were then dot blotted to the PVDF membrane (Immobilon P, Millipore, St. Louis, MO, USA), and acidic glycoproteins were visualized with Alcian Blue (see Appendix A).

The PVDF membrane spots were excised and placed in test tubes (two spots/sample), followed by 5 × 15 min destain/washes in MeOH. The glycans were released from the protein with a 40 µL beta elimination solution (0.5 M NaBH_4_ in 0.05 M NaOH) at 50 °C in a water bath. The samples were neutralized with 1–2 μL conc HAc, followed by desalting using cation exchange media (AG50WX8 (Biorad, Hercules, CA, USA)) in C18 ziptips (Millipore), two ziptips/sample, and dried with speedvac. Borate residuals were eliminated by repeated additions of MeOH (5 × 50 μL) and evaporated in between.

#### 4.4.2. Glycan Analyses with LC/MS

Reduced glycans were resuspended in 6 µL of water and injected (2 μL) onto a liquid chromatography–electrospray ionization tandem mass spectrometry (LC-ESI/MS). The oligosaccharides were separated on a column (10 cm × 250 µm) packed in-house with 5 µm porous graphite particles (PGC, Hypercarb, Thermo-Hypersil, Runcorn, UK) and a flow rate of 5 μL/min. The oligosaccharides were eluted with the following gradient: 0–46 min 0–45% B, wash 46–54 min 100% B, then equilibration between 54–78 min with 0% B. Buffer A was 10 mM ammonium bicarbonate (ABC) and buffer B was 10 mM ABC in 80% acetonitrile. 

A 30 cm × 50 µm i.d. fused silica capillary was used as a transfer line to the ion source. The samples were analyzed in the negative ion mode on an LTQ linear ion trap mass spectrometer (Velos Pro, Thermo Electron, San José, CA, USA), with an IonMax standard ESI source equipped with a stainless-steel needle kept at −2.5 kV. Compressed air was used as nebulizer gas. The heated capillary was kept at 270 °C. A full scan (*m*/*z* 380–1800, two microscan, maximum 100 ms, target value of 30,000) was performed, followed by data-dependent MS^2^ scans (two microscans, maximum 100 ms, target value of 10,000) with a normalized collision energy of 35%, isolation window of 3 units, activation q = 0.25, and activation time of 30 ms). The threshold for MS^2^ was set to 300 counts. Data acquisition was conducted with the Xcalibur software (Version 2.0.7). 

#### 4.4.3. MS^2^ Spectra Interpretation

The obtained tandem mass spectrometry (MS/MS) spectra were interpreted manually and confirmed using the freely available software “GlycoWorkbench” [59]. Since the species analyzed in this project have not been characterized previously, interpretations are just based on similarities to already characterized glycans. The spectra were compared to structures from humans and mice, stored in the Unicarb-DB database (www.expasy.org (accessed on 1 January 2023)) when available, and also compared to reference spectra from mucin glycan interpretations from Atlantic salmon [14]. Peak quantification was performed manually using the Xcalibur software (Thermo Scientific). Note that the MS ionization efficiency for individual glycans may vary slightly, because, for example, acidic glycans may ionize better than neutral glycans in a negative ion mode. MS fragmentation cannot distinguish between different hexoses and *N*-acetylhexosamines. The monosaccharide symbols used in the figures follow the SNFG (Symbol Nomenclature for Glycans) symbols. Supportive evidence for typical core 2 branching (R-Galβ1-3(R-GlcNAcβ1-6)GalNAc-Ser/Thr) is obtained via the diagnostic ion ^0,2^A_0_ [60]. This arises from cleavage between C-3 and C-4 in the GalNAc residue that is linked to the peptide backbone and is annotated as ^0,2^A_0_ in Figure 5.

#### 4.4.4. Chemicals

Chemicals were from Sigma-Aldrich unless stated otherwise. 

#### 4.4.5. Pilot Experiments

Aliquots of one sample from each of the three different elasmobranch species were prepared and analyzed twice at increasing amounts in initial pilot experiments using a standard glycomics workflow (see Methods). Since no glycans were detected, a third pilot experiment was performed by pooling five samples from spiny dogfish. Increasing the starting amounts allowed for the detection of 29 *O*-glycans and 3 *N*-glycans. The BCA assay (Appendix A) results for protein concentration far exceeded the levels normally required for glycomics; however, we could use this as a guideline to pool and process the remaining samples (Appendix A). In hindsight, the BCA protein assay most likely did not truly reflect the glycoprotein content, since hardly any glycans were detected in the catshark samples.

## 5. Conclusions

This is the first study that comprehensively examines the mucus layer of two shark and one skate species including histology and glycoproteomics. Several novel glycans were identified that differ to glycans previously observed in teleost fish, and some bear resemblance to human glycans. While speculative, it may be that since shark skin is covered with denticles that reduce drag there is less of a need for a thick mucus layer. Conversely, it may also be that various molecules, for example, antimicrobial molecules, are more concentrated or more potent in the shark thin mucus layer. Further research on elasmobranch skin is needed, especially bioprospecting studies that aim to identify novel molecules, understand their function, and, if possible, translate them to human clinical use. 

## Figures and Tables

**Figure 1 ijms-24-14331-f001:**
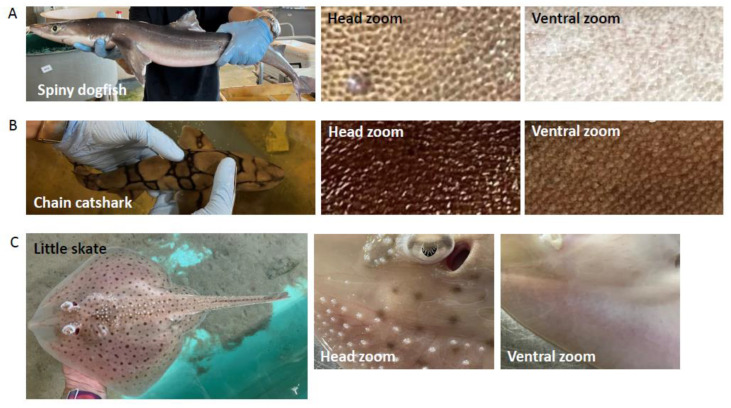
Sharks and skates studied. Photos of spiny dogfish *(Squalus acanthias*) (**A**), chain catsharks (*Scyliorhinus retifer*) (**B**), and little skate (*Leucoraja erinacea*) (**C**) at the Marine Resources Center, Marine Biological Laboratory, Woods Hole. In the head and ventral zoom photos, the placoid scales typical for elasmobranchs are visible.

**Figure 2 ijms-24-14331-f002:**
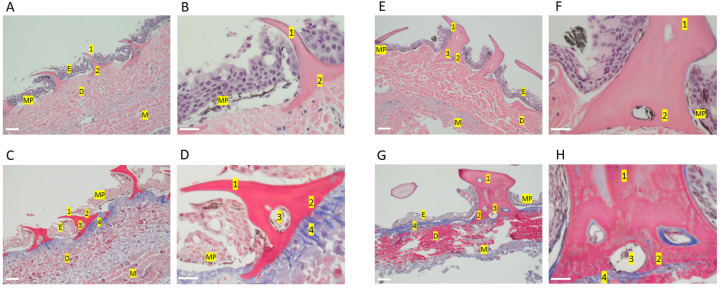
Skin histology of spiny dogfish (**A**–**D**) and chain catsharks (**E**–**H**). All images are sagittal sections of skin biopsy. Representative images of one shark from each species is shown. H-E staining (**A**,**B**,**E**,**F**) shows the different skin layers stated: epidermis E, dermis D, and denticles 1,2. MT staining (**C**,**D**,**G**,**H**) shows a more refined division of the skin layers and allows the differentiation of hard tissues (e.g., teeth, denticles) from soft ones; collagen appears blue and keratin and muscle fibers red. Layers are indicated as follows: epidermis (E), dermis (D), muscle (M), melanin pigmentation (MP), denticles 1,2. Denticles consist of a backward-pointing spine (1), a basal plate covered with enamel (2), and a pulp cavity (3). Blue (4) indicates collagen. Images were taken at 10× ((**A**,**C**,**E**,**G**) 100 μm) and 40× ((**B**,**D**,**F**,**H**) 50 μm).

**Figure 3 ijms-24-14331-f003:**
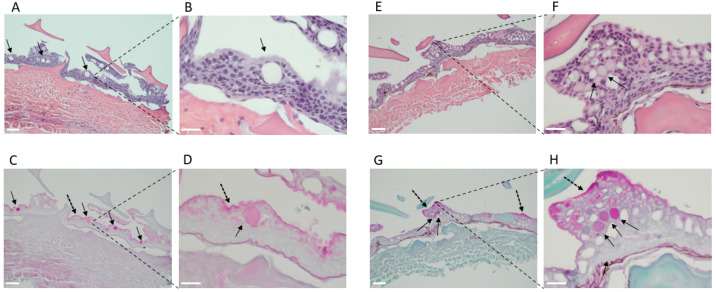
PAS staining in shark skin demonstrates the presence of mucin secretory cells—overview. Spiny dogfish (**A**–**D**) and chain catsharks (**E**–**H**). All images are sagittal sections of skin biopsies. Representative images of one shark from each species are shown. H-E staining (**A**,**B**,**E**,**F**) shows the location of the mucin vacuoles throughout the epidermis layer (in purple). PAS staining (**C**,**D**,**G**,**H**) shows the mucosal layer in magenta-pink at the apical part of the epidermis (black dotted arrow). Glycoproteins (including mucins) and glycolipids are shown in pink-magenta- and light-blue-colored vacuoles (black arrows). Images were taken at 10× ((**A**,**C**,**E**,**G**) 100 μm) and 40× ((**B**,**D**,**F**,**H**) 50 μm).

**Figure 4 ijms-24-14331-f004:**
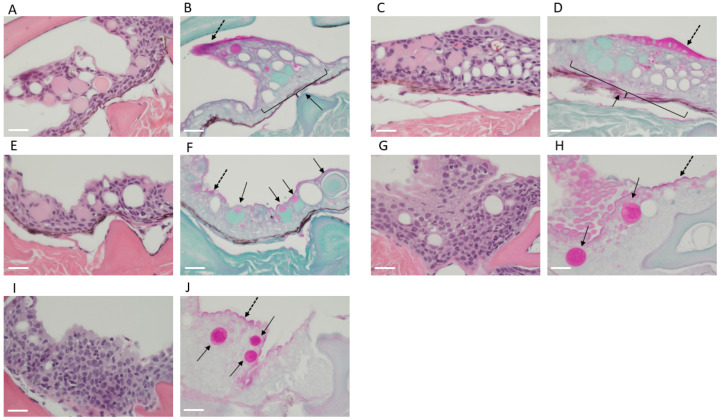
PAS staining in shark skin demonstrates the presence of mucin secretory cells—zoomed-in images of (**A**–**F**) chain catsharks and (**G**–**J**) spiny dogfish. All images are sagittal sections of skin biopsies. Representative images of three catsharks and two dogfish are shown. H-E staining (**A**,**C**,**E**,**G**,**I**) shows the location of the mucin vacuoles throughout the epidermis layer (in purple). And the equivalent PAS staining (**B**,**D**,**F**,**H**,**J**) shows the mucosal layer in magenta-pink at the apical part of the epidermis (black dotted arrow). Glycoproteins (including mucins) and glycolipids are shown in pink-magenta- and light-blue-colored vacuoles (black arrows). All images were taken at 40 × 50 μm.

**Figure 6 ijms-24-14331-f006:**
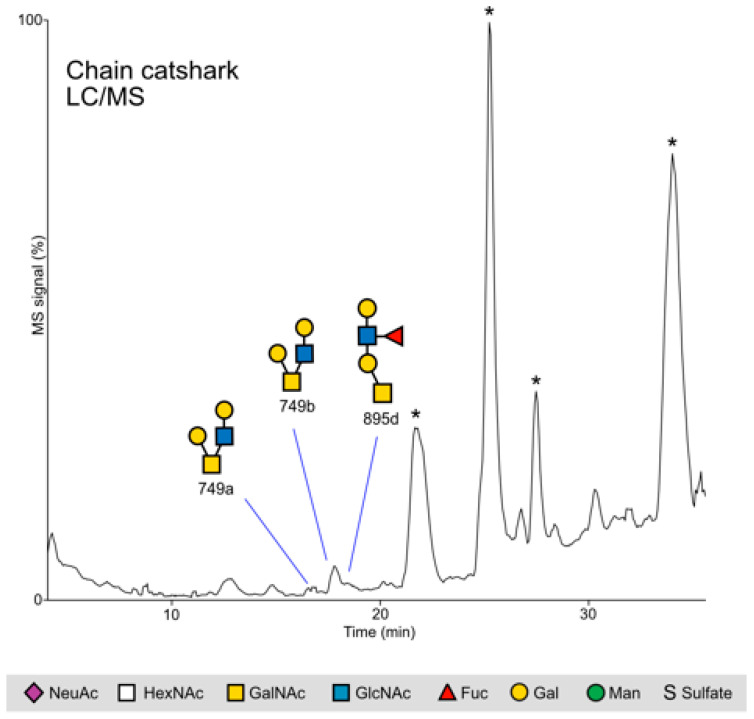
Catshark glycans. *O*-glycans were analyzed in their reduced nonderivatized form using LC/MS in the negative ion mode. * = non glycan contaminant. Note that other sampling methods may increase the number of glycans identified.

**Figure 7 ijms-24-14331-f007:**
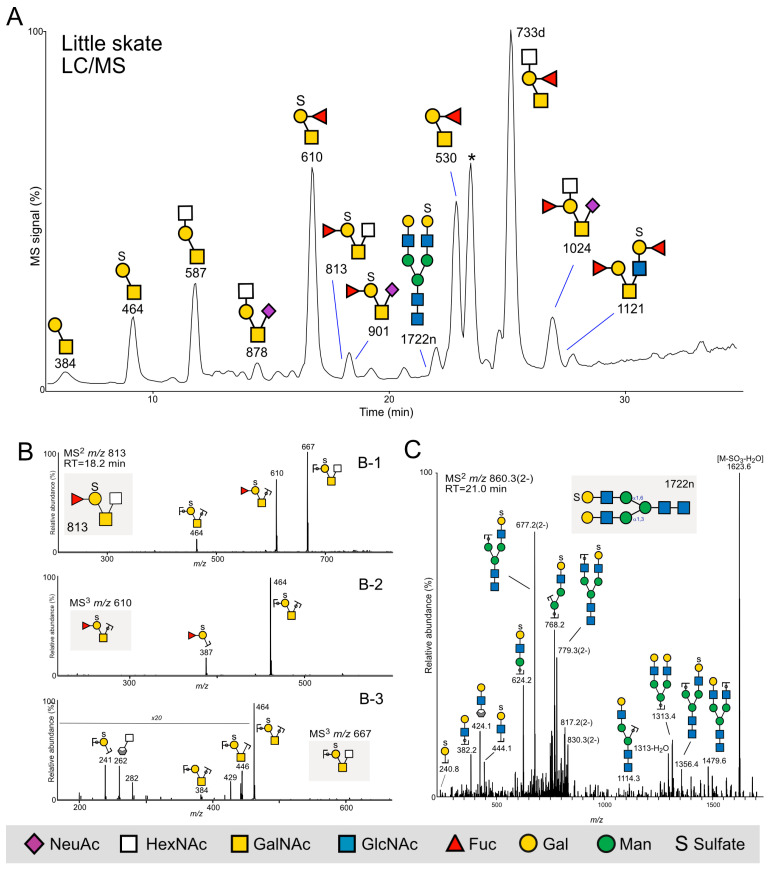
Little skate glycans. (**A**) *O*-glycans from two pooled samples were analyzed in their reduced nonderivatized form using LC/MS in the negative ion mode. * = nonglycan contaminant (**B**) MS^2^ and MS^3^ experiments of pooled samples from “A” shows a glycan detected at *m*/*z* 813 and eluting at 18.2 min. MS^3^ fragmentation experiments of the fragment ions at *m*/*z* 610 and 667 from MS^2^ supported the sequence assignment. (**C**) MS^2^ spectra of a sulfated *N*-glycan (“1722n”, Appendix A), eluting at 21.1 min and detected as a doubly charged ion at *m*/*z* 860.3^2−^ (precursor ion [M-2H]^2−^).

**Figure 8 ijms-24-14331-f008:**
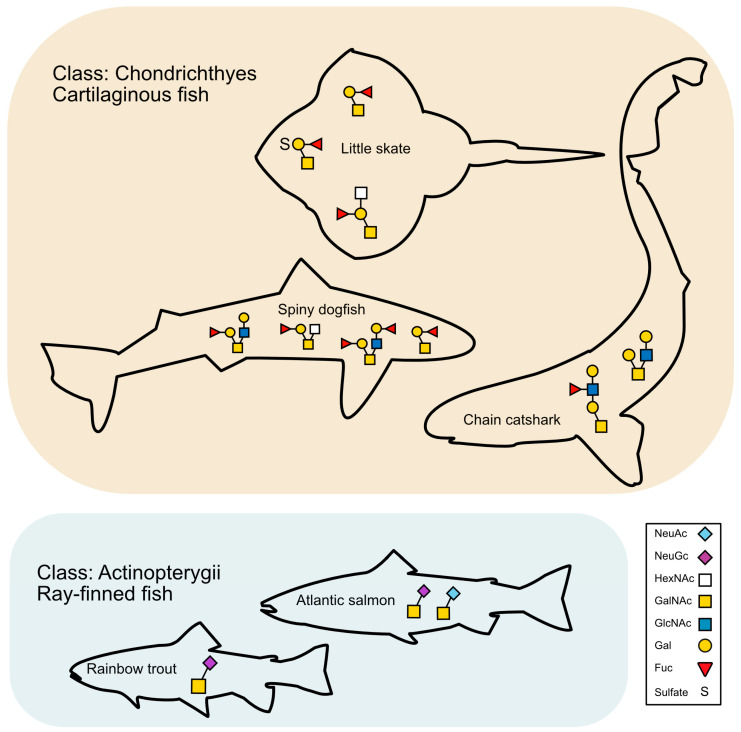
Summary of the most common elasmobranch skin glycans and comparison with the most common previously described skin glycans from salmonids. A cartoon of the main glycans that together constitute >50% of total glycans (based upon MS signal response) in spiny dogfish (51%), chain catshark (100%), and little skate (74%) compared with Atlantic salmon (74%) and rainbow trout (81%) [15,22]. Little skate: *m*/*z*: 530, glycan name: Fuc(a1-2)Gal(b1-3)GalNAcol, *m*/*z* 733, glycan name: Fuc(a1-2)(HexNAc-)Gal(b1-3)GalNAcol, *m*/*z* 610 glycan name: Fuc(a1-2)(SO_3_^−^)Gal(b1-)-GalNAcol. (Scheme 1041. b,): glycan name: Fuc(a1-2)Gal(b1-3)[Fuc(a1-2)Gal(b1-3)GlcNAc(b1-6)]GalNAcol, *m*/*z*: 530 glycan name: Fuc(a1-2)Gal(b1-3)GalNAcol. *m*/*z* 733, b, glycan name: Fuc(a1-2)Gal(b1-3)[HexNAc(b1-6)]GalNAcol and *m*/*z*: 895, c, glycan name:, Fuc(a1-2)Gal(b1-3)[Gal(b1-4)GlcNAc(b1-6)]GalNAcol, respectively. Chain catsharks: *m*/*z*: 749a,b, glycan name: Hex-(Hex-HexNAc-)HexNAcol, Gal(b1-3)[Gal-GlcNAc(b1-6)]GalNAcol, respectively, and *m*/*z* 895d glycan name: Hex-(deHex-)HexNAc-Hex-HexNAcol. In both fish shown, there are short NeuAc-containing *O*-glycans, the most abundant being the disaccharide NeuAcα2-6GalNAc.

## Data Availability

Not applicable.

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
