# Peer review of "Identification of Novel Glycans in the Mucus Layer of Shark and Skate Skin"

_ijms, 2023, doi:10.3390/ijms241814331_

Round 1
Reviewer 1 Report
The study aimed to explore shark and skate skin mucus composition and glycoproteins.
1.Histology of spiny dogfish and chain catshark skin showed mucin-producing goblet cells and a mucus layer.
2.Liquid chromatography-mass spectrometry identified diverse O-glycans and some N-glycans in spiny dogfish, chain catshark, and little skate skin mucus.
3.Shark mucus O-glycans differ from known teleost fish mucus O-glycans by being more sulfated and fucosylated.
4.Several shark mucus glycans resemble human gastric and blood group H glycans.
5.Spiny dogfish mucus had mostly neutral fucosylated core 1 and core 2 O-glycans.
6. Skate mucus had some acidic and sulfated O-glycans not seen in the sharks.
The findings provide a basis for further research on elasmobranch skin mucus function and potential biomedical roles.
The author in this article provides a lot of data to support Identification of novel glycans in the mucus layer of shark and. skate skin. But I have a few questions about this animal skin article.
1). The clarity in the picture is not enough, is it because of the typesetting problem? When typesetting, we should actively communicate with the editor in ijms to upload TIFF or high-quality JPG. This problem is particularly strong in Figures 5 and 8, and it is not known what the meaning of the message is conveyed without the remark.
2)Figure 3E is clearly a screenshot. Can you change it to a normal picture?
3) Why don't all the pictures have a Scale Bar?
4)In general, I like this article very much, because this article has a good significance for the development of dermatology, and the author has spent a lot of energy in collecting samples. It also fully shows the diversity of ijms. However, this article expresses novel glycans in the bottom layer of shark and skate skin. If you can add some Immunolgy staining(IHC) or do some biochemical experiments such as WB to explain the Identification of novel glycans in the bottom layer of shark and skate skin, it is the best. For example, is mucin secretory cells marker more convincing to stain shark and skate skin? Be aware that ijms are biological articles, and biological experiments cannot just stop at the tissue level. Identification of novel glycans using LC/MS is a very excellent way. Thanks to the author for giving us who study dermatology some different vision.
Minor editing of English language required
Author Response
1) The clarity in the picture is not enough, is it because of the typesetting problem? When typesetting, we should actively communicate with the editor in ijms to upload TIFF or high-quality JPG. This problem is particularly strong in Figures 5 and 8, and it is not known what the meaning of the message is conveyed without the remark.
Answer: This is a typesetting issue from the journal. In the uploaded powerpoint file the images are sharp.
2)Figure 3E is clearly a screenshot. Can you change it to a normal picture?
Answer: This is a typesetting issue from the journal. In the uploaded powerpoint file the images are sharp.
3) Why don't all the pictures have a Scale Bar?
Answer: Thanks for pointing this out, scale bars have been added.
4)In general, I like this article very much, because this article has a good significance for the development of dermatology, and the author has spent a lot of energy in collecting samples. It also fully shows the diversity of ijms. However, this article expresses novel glycans in the bottom layer of shark and skate skin. If you can add some Immunolgy staining(IHC) or do some biochemical experiments such as WB to explain the Identification of novel glycans in the bottom layer of shark and skate skin, it is the best. For example, is mucin secretory cells marker more convincing to stain shark and skate skin? Be aware that ijms are biological articles, and biological experiments cannot just stop at the tissue level. Identification of novel glycans using LC/MS is a very excellent way. Thanks to the author for giving us who study dermatology some different vision.
Answer: We understand that more molecular experiments are needed to decipher the roles of shark glycans however this is a pilot study on hard to access specimens in which access to material is limited. The study was conducted during a 9 week field study away from our home institute. The mucus material used for the mass spectrometry has been fully used and none remains for additional experiments. Regarding WB experiments this are very difficult to perform on rare shark species because of lack of antibodies as well as material.
Minor editing of English language required
Answer: We have gone through the language to the best of our abilities.
Reviewer 2 Report
Dear authors,
The manuscript titled " Identification of novel glycans in the mucus layer of shark and 2 skate skin” by Etty Bachar-Wikstrom et al. is a study that comprehensively examines the mucus layer of two shark and one skate species including histology and glycoproteomics. The text is well structured, clearly presented in methods and results, and well-discussed. These data are the basis for future studies and applications in translational medicine.
Minor point:
I suggest that the authors insert the data of the "supplementary files" in the main text
Author Response
I suggest that the authors insert the data of the "supplementary files" in the main text
Answer: We thank the reviewer for the positive outlook on the manuscript. We understand the rational for moving the supplemental figures to the main text however we feel these figures are more appropriate as supplementary as they are supporting information and do not belong to the main results of the manuscript.
Round 2
Reviewer 1 Report
The most attractive to me is histological examination of skin samples from Atlantic spiny dogfish (Squalus acanthias) sharks and chain catsharks (Scyliorhinus retifer) revealed distinct mucin-producing cells and a mucus layer, indicating the presence of a functional mucus layer similar to bony fish mucus albeit thinner.If there is a chance to find more antibodies to contribute to us.I think it can be published.